# Nutritional and Physicochemical Characterization of *Strychnos madagascariensis* Poir (Black Monkey Orange) Seeds as a Potential Food Source

**DOI:** 10.3390/foods9081060

**Published:** 2020-08-05

**Authors:** Kiana Kirsty van Rayne, Oluwafemi Ayodeji Adebo, Nomali Ziphorah Ngobese

**Affiliations:** 1Department of Botany and Plant Biotechnology, University of Johannesburg, P.O. BOX 524, Johannesburg 2006, South Africa; kianakvanrayne@gmail.com; 2Department of Biotechnology and Food Technology, University of Johannesburg, P.O. BOX 17011, Johannesburg 2028, South Africa; oadebo@uj.ac.za

**Keywords:** colour, FTIR, macronutrients, micronutrients, strychnine, texture

## Abstract

*Strychnos madagascariensis* Poir is an underutilized fruit that is considered a valuable food during droughts and famine. The aim of this research was to characterize the nutritional composition and the flour functional properties, for the use as a potential food source. Seed flour was analysed using a standard enzymatic assay for sugars, acid/neutral detergent analysis for fibre, ether extraction for fat and HPLC for strychnine. Results showed that the seeds contained 41% reducing sugars and 53% fibre. The mineral composition, determined using microwave-assisted acid digestion and inductively coupled plasma optical emission spectrometry (ICP–OES), showed that the seeds contained high quantities of iron (15.78 mg/100 g) and manganese (9.86 mg/100 g). The flour water absorption index (1.37 g/g) was substantially higher than that of wheat, brown rice and tapioca flours and the oil absorption index showed similarities to the reference flours (1.09 g/g). The flour peak (37,788 RVU) and final viscosities (62,928 RVU) were significantly (*p* < 0.001) higher than the reference flours. This study was the first to quantify the strychnine content (0.08%) in the seeds. Results suggest that the seeds have good potential for food product development; however, further processing is essential to ensure safety for consumption.

## 1. Introduction

Many developing countries are faced with the problem of food insecurity, which is threatening the quality of life of inhabitants. This issue is also on the rise as the world population is rapidly increasing, placing a higher demand on the currently available food staples and imported food goods [1]. Thus, native edible plants as staple alternatives or supplements may be used to combat this problem, by lessening the demand on current staples and reducing the need for food importation [1,2]. According to Van Wyk [3], South Africa is home to a large variety of edible plants that may be used to produce food and beverages, for both the local and international market.

Coupled with the threat of climate change and a drastic reduction in rainfall (particularly in southern Africa), agricultural food production has been greatly affected, thus worsening the food security status and increasing the dependence on imported staple foods. It has been proposed that the use of underutilized indigenous food crops may improve food production as certain plants are well adapted to the environment and require substantially low amounts of water, low agronomical inputs, as well as other characteristics that allow them to adapt to current climate changes. Moreover, deviation from the current agricultural mono-cropping systems and the introduction of a larger variety of underutilized crops may contribute greatly in the adoption of agro-biodiversity, which benefits the environment [4].

A rising interest has developed in South Africa, around the genus *Strychnos* (Monkey orange) as a new food crop due to specific species within this genus. These species are known for their production of edible fruits and high drought tolerance, making them suitable for arid regions [5]. Particularly, *S. madagascariensis* Poir (commonly known as black monkey orange) belongs to the family *Loganiaceae*. It is a native southern African tree, which is distributed in Limpopo, North-West, Mpumalanga, Eswatini, KwaZulu-Natal and Botswana. The tree produces high fruit yields, containing numerous large translucent seeds. According to Shaffer [2], only the fruit pulp of *S. madagascariensis* is consumed (ranging between bitter to very sweet in taste) as it is believed that the seeds may contain the toxic alkaloid, strychnine. However, there has been contradictory claims surrounding the toxicity of the seeds as Govender [5] reported that the seeds are consumed. The toxicity of the seeds has not been proven and is not supported by any ethnobotanical data. The fruit is considered to be an underutilized commodity as its full potential as a food source has not yet been realized. Underutilized food plants typically provide higher nutritional value in comparison to those commonly consumed [6]. Thus, the aim of this study is to determine the nutritional composition of these seeds, characterize the functional properties of the seed flour and to determine its safety as a potential food source.

## 2. Materials and Methods 

### 2.1. Preparation of Flour Samples

Ripe (orange) fruits of *S. madagascariensis* were obtained from the Agricultural Research Council (ARC) in Nelspruit, South Africa (25°27′04.6″ S, 22°56′15.0216″ E). The seeds were removed from the fruit pulp and freeze-dried for five days. The seeds were then milled and further ground using a coffee grinder (Bosch, South Africa). The ground material was then passed through a 250 µm sieve and stored in airtight zip-lock plastic bags. Reference flours were purchased at local supermarkets: uncooked, gluten-enriched wheat flour (Premier, South Africa), long-grain gluten free brown rice flour (Health Connection Wholefoods, South Africa), dehulled split chickpea flour (Natures Choice, South Africa), coconut flour (Health Connection Wholefoods, Philippines) and tapioca flour (Health Connection Wholefood, Thailand).

### 2.2. Macronutrient Determination

The moisture content of the flour samples was determined using the MLB 50-3N moisture meter (Kern, Germany). The total fibre content of the seeds was calculated using the acid detergent fibre (ADF) fraction and the neutral detergent fibre (NDF) fraction. These components were determined using the methods set out by Goering and Van Soest [7], in which samples were digested in an acid-detergent solution and a neutral-detergent solution (respectively) using a digestion block. Reducing sugars and starch quantification were performed using the Somogyi–Nelson assay and spectrophotometry methods set out by Al-Mhanna [8]. The fat content was determined through ether extraction, using the methods of AOAC [9]. The protein fraction was determined by multiplying the total nitrogen composition (obtained during ICP–OES mineral analysis) with the conversion factor (6.25) to yield the crude protein composition. The reference samples were obtained from the South African foods data system [10]. The total carbohydrate fraction was calculated using the sugar, fibre and starch compositions. The total energy was then calculated based on the macronutrient composition using Equation (1), excluding the contribution of the fibre content in the carbohydrate fraction. The percentage contribution of the flour to the nutrient reference values (NRVs) were calculated using Equation (2). The NRVs for macronutrients were obtained from the European Food Safety Authority [11] and the World Health Organization [12]:Kcal = (% protein × 4) + (% carbohydrate × 4) + (% Fat × 9)(1)
% NRV = (Nutrient/NRV) × 100(2)

### 2.3. Mineral Analysis

The mineral composition (Na, P, Ca, Mg, K, Cu, Fe, Zn, Mn, N) was determined according to Santos [13], with slight modifications. The samples were prepared using microwave-assisted acid digestion. Approximately 500 mg of the sample was placed into the vessel together with 10 mL of 65% nitric acid. The vessels containing the sample and the standard solutions were then placed into the microwave, model MARS 6240/50 (Matthews, NC, USA). The digestion was run using the plant material setting with a ramp time of 20–25 min, at 200 °C and a hold time of 10 min. The pressure within the vessels was released and the digested plant material was transferred into 10 mL falcon tubes. Analysis was performed using ICP-OES, with the conditions for radiofrequency power (1.3 kW), nebulizer pressure (150 kPa), plasma argon flow rate (15 L/min) and auxiliary argon flow rate (1.5 L/min). The percentage contribution of the micronutrient composition of the flour was calculated using Equation (1), mentioned above. The NRVs for the macronutrients were obtained from the Expert Group on Vitamins and Minerals [14].

### 2.4. Quantification of Strychnine 

Strychnine was quantified using high performance liquid chromatography (HPLC), UV detection (254 nm), 25 cm × 4.2 mm ID column (packed with μBondpack C18, 10 μm particle size) and the methods outlined in the NIOSH (National Institute for Occupational Safety and Health) Manual of Analytical Methods (NMAM) [15]. Strychnine was extracted from the flour sample (1 g) using 5 mL of the mobile phase (1.1014 g of 1-heptanesulfonic acid sodium salt in 980 mL of 1:1 acetonitrile (chromatographic quality): water made up to 1 L, pH 3.5). The solution was then filtered through a 0.2 µm syringe filter and placed in an autosampler. The quantity of strychnine was determined through the use of a standard curve constructed using standard solutions of strychnine (Sigma Aldrich, Germany).

### 2.5. Gas Chromatography Mass Spectrometry (GC–MS) Analysis

Sample preparation for metabolite profiling was done according to the methods set out by Adebo [16] with slight modifications. 3 g of the flour samples were weighed into centrifuge tubes and mixed with 30 mL of 80% aqueous methanol (Merck, South Africa). This solution was agitated and incubated for 24 h on a shaking incubator at 4 °C. Following the incubation, the solutions were centrifuged at 3500 rpm at 4 °C for 5 min. The supernatant was then concentrated in a round bottom flask using a rotavapor under vacuum at 40 °C. The concentrated extract was filtered through a 2 µm syringe filter and transferred into dark amber vials. The extraction was done in triplicate for each flour sample and the analysis was done on a gas chromatography high resolution time of flight mass spectrometry (GC-HRTOF-MS) system having an Agilent 7890A gas chromatograph (Agilent Technologies, Inc., Wilmington, DE, USA) operating in high resolution, with a Gerstel MPS multipurpose autosampler (Gerstel Inc. Germany) and a Rxi^®^-5 ms column (30 m × 0.25 mm ID × 0.25 μm) (Restek, Bellefonte, USA). 1 µL was injected in splitless mode with helium as the carrier gas. The initial oven temperature was set at 70 °C, held for 0.5 min, ramped at 10 °C/min to 150 °C, held for 2 min, ramped at 10 °C/min to 330 °C and held for 3 min. Subsequent data obtained were processed by evaluating each of the three replicates and selecting only the compounds found within each replicate. Once this was done, the common compounds found within *S. madagascariensis* seed flour were compared to those of the reference flours to construct a table of comparison.

### 2.6. Water and Oil Absorption Index

The water absorption index (WAI) and the oil absorption index (OAI) of the flour samples were determined using the methods set out by Julianti [17]. Approximately 1 g of each flour sample was placed into centrifuge tubes, together with 10 mL of distilled water (for WAI) or 10 mL of sunflower oil (Pick n Pay Retailers, South Africa) for OAI. The suspensions were vortexed for 1 min and centrifuged at 1000× *g* for 10 min at 25 °C. The supernatant was then discarded, and the pellet was weighed. The WAI and OAI were calculated using the formula:WAI or OAI = (Weight of pellet)/(Weight of flour sample)(3)

### 2.7. Swelling Power and Solubility

The swelling power (SP) and the solubility (S) of *S. madagascariensis* seed flour and the reference flours were determined using the methods of Yu [18] with minor modifications. Approximately 1 g of the sample was placed into centrifuge tubes together with 15 mL of distilled water. A slurry was made by vortexing the mixture until all the powder was dissolved. The solution was then heated in a water bath (Optolabor, Johannesburg) at 50, 60, 70, 80, 90 and 100 °C for 30 min. The tubes were allowed to cool to room temperature and then centrifuged at 2600× *g* for 15 min. The supernatant was decanted into its respective pre-weighed glass petri dishes and evaporated at 105 °C, for 24 h. They were allowed to cool to room temperature and the resulting pellets weighed. Solubility and swelling power were calculated using Equations (3) and (4), respectively:S = Wr/W × 100(4)
SP = Wt/(W − Wr)(5)where Wr is the weight of the dried supernatant; W is the weight of the sample and Wt is the weight of the wet sediment (pellet).

### 2.8. Fourier Transform Infrared (FTIR) Spectroscopy 

FTIR spectroscopy has gained popularity in the food industry as it is a rapid and non-destructive technique for the analysis of food compositions. This technique is simple and requires little to no sample preparation as samples may be analysed in their original state [19]. The IR Affinity-1S series FTIR equipment (Shimadza, Japan) was used for the analysis. The method of Sujka [20] was followed for the analysis. The flours were placed in a measuring chamber and were scanned within the spectral range of 4000–370 cm^−1^ using a resolution of 4 cm^−1^.

### 2.9. Pasting Properties

The pasting properties of the test and reference flours were determined using a rapid visco analyser (RVA) (4500, Perten Instruments, Sweden) following the methods of Kongolo [21]. Suspensions were made containing flour and distilled water (14% *w*/*w*, flour in dH_2_O). The suspensions were then equilibrated for 1 min at 50 °C and stirred for 30 s at 960 rpm. The stirring speed was then reduced to 160 rpm and the temperature was increased to 91 °C at a rate of 11.08 °C/min. Once the samples had reached the desired temperature, they were held at that temperature for a further 2.5 min. The temperature was then decreased to 50 °C and held for 1 min, after which the critical points (pasting temperature, peak viscosity, final viscosity, breakdown viscosity and set back viscosity) were determined.

### 2.10. Gel Texture Properties

The gel texture properties of the test and reference flours were analysed using a TA.XT Plus Texture Analyzer (Stable Micro Systems, Brighton, UK) and a 2 mm cylinder probe. The pastes produced in Section 2.9 were used for this analysis. The pastes were transferred into plastic sample holders (dimensions: 2 cm in height and 3.5 cm wide) and allowed to set for 2 h at 25 °C. The texture analyser was programmed using the following settings: pre-test and test speed—1.0 mm/s, post-test speed—10.0 mm/s and penetration distance—5 mm. The texture properties (gel firmness, penetration energy) were determined using the TA.XT Exponent© software.

### 2.11. Colour Properties

The colour properties of the test and reference flours were determined using the tristimulus Commission Internationale de l’Eclairage’s (CIELAB) scale, using a CR-10 chromameter (Konica Minolta Sensing Inc., Tokyo, Japan). The CIELAB values L * (white = 100 and black = 0), a * (positive = red and negative = green) and b * (positive = yellow and negative = blue) were recorded. The chroma, yellowness and flour colour index (FCI) were calculated using Equations (6)–(8), respectively:Chroma = √(a˄2 + b˄2)(6)
Yellowness = (142.86 × b)/L(7)
FCI = L − b(8)

### 2.12. Statistical Analysis

Except for FTIR, all other analyses were done in triplicate. Genstat for Windows 19th Edition (VSNI, UK) was used for statistical analyses. A one-way analysis of variance (ANOVA) and Tukey’s post hoc test was performed to differentiate between the means at the 95% probability level.

## 3. Results and Discussion

### 3.1. Nutritional Evaluation

#### 3.1.1. Micro and Macronutrient Composition

When analysing the macronutrient composition of the seed four (Table 1), the carbohydrate fraction (~89.85%) was noted as the major component of the seeds; furthermore, this fraction is predominantly composed of fibre (~53%) and reducing sugars (~41%). The other macronutrient fractions of the seed flour, such as fats (~0.95%, similar to that of wheat flour, 1%) and protein (~8.27%, similar to that of wheat flour, 8.2%) constitutes less than 10% of the flour composition. Furthermore, it was found that the seeds contain no starch. The fibre content of *S. madagascariensis* seeds is far greater than that of dried broad beans (22.9%), a popular fibre-rich food [22]. According to Schweizer and Würsch [23] and Maćkowiak [24], fibre has beneficial effects in the gastrointestinal tract. A high prebiotic fibre diet improves the functioning of the gastrointestinal tract as it acts as a substrate (prebiotic) for the build-up of beneficial microflora and reduces the risk of constipation. Furthermore, the improved functioning of the gastrointestinal tract, assisted by the fermentation of prebiotic dietary fibre, aids in digestion, by promoting the uptake of minerals [25], thus further highlighting the potential of *S. madagascariensis* seeds for the possible alleviation of mineral deficiencies. However, further investigation is required to quantify the amount of prebiotic dietary fibre present in the seeds. *S. madagascariensis* seeds contain moderate amounts of energy (189.03 kcal), which is substantially lower than that of commercial flours (i.e., wheat flour 358 kcal, brown rice flour 365 kcal and chickpea flour 378 kcal, data which were obtained from the food composition tables of South Africa [22]). Although the energy content is lower than that of commercial flours, the seeds may still hold potential as a valuable source of nutrients.

When comparing the micronutrient composition (Table 1) of the seeds to that of popular flours and mineral rich foods found in the food composition tables of South Africa [22], it is observed that the Ca (148.03 mg/100 g), Fe (15.78 mg/100 g), K (594.36 mg/100 g) and Mn (9.86 mg/100 g) are present in substantially high quantities. In particular, the Ca and Fe contents of *S. madagascariensis* seeds were higher than those found in lentil (Ca = 80 mg/100 g and Fe = 6.9 mg/100 g), wheat (Ca = 14 mg/100 g and Fe = 1.2 mg/100 g) and brown rice flour (Ca = 11 mg/100 g and Fe = 2 mg/100 g). The K content was found to be substantially higher than that reported for peeled bananas (241 mg/100 g) wheat flour (105 mg/100 g) and brown rice flour (289 mg/100 g), while the Mn content greatly exceeded that found within wheat (0.63 mg/100 g) and brown rice flour (4 mg/100 g). The seeds also contained Mg (79.25 mg), P (94.20 mg), Na (16.45 mg), Zn (1.36 mg) and Cu (0.72 mg). The rich mineral composition thus suggests that the use of the seed of *S. madagascariensis* may have the potential in alleviating some mineral deficiencies associated with developing/underdeveloped countries. Moreover, Fe deficiency is considered to be the most predominant micronutrient deficiency across the world [26]. In southern Africa, it is reported that females and black Africans are most susceptible to Fe deficiency [27].

Many species within the genus *Strychnos* are considered to be toxic due to the presence of strychnine. The strychnine content of *S. madagascariensis* seeds (0.08%) was found to be substantially lower than that of *S. nux-vomica* (~1.1%) [28]. However, the seeds may still be harmful if consumed. Furthermore, the processing of these seeds may further reduce or eliminate the strychnine content. Further investigation is required to evaluate the strychnine content after processing. *Strychnos nux-vomica* is the main member of the genus that has made it popular—it is cultivated in many parts of the world and is a medicinally important plant, specifically in Indian and Chinese remedies. Although the seeds are the main target, the leaves, bark and fruit are utilized for herbal remedies as well after they have been processed (boiled or cooked) in order to reduce the quantity of strychnine. These techniques might not be effective in reliably reducing the strychnine content [29,30], nor being suitable in the development of a potential food product. As such, other conventional and novel food processing techniques should be considered to effectively reduce the strychnine content, to improve its food use. Traditionally, the seeds of *S. nux-vomica* are used for an array of therapeutic and clinical applications, which include increasing appetite, reducing fevers, and being used as an aphrodisiac and a tonic [31]. Furthermore, a study conducted by Chen et al. [28] revealed that the processing (treated with 50% ethanol) of the seed powder of *S. nux-vomica* significantly improved the medicinal efficiency and greatly reduced the toxicity, in comparison to the unprocessed seed powder. Another study conducted by Choi et al. [32] evaluated the traditional processing techniques of *S. nux-vomica* seeds, and reported an effective reduction in the strychnine content, among other alkaloids. These studies, although developed for the production of medicine and herbal remedies, reveal that strychnine may be substantially reduced. Further research is required to develop methods suitable for food product development, to efficiently eliminate the strychnine content within *S. madagascariensis* seeds and ensure safety for consumption.

#### 3.1.2. GC–MS Analysis

Table 2 documents those compounds found within *S. madagascariensis* seed flour. The seed flour shared some compounds with the reference flours; however, the majority of the compounds identified were unique to *S. madagascariensis*. The compounds shared between the seed powder and reference flours include pyrazole, vinyl acrylate, ethylene acetal and methoxy-phenyl-oxime. To ensure that the seed flour of *S. madagascariensis* is safe for consumption and food product development, further analysis is essential to quantify the compounds listed in Table 2, as compounds such as vinyl acrylate, trifluoromethylcinnamic acid, methylcyclopentenolone and silicon tetrafluoride may exist in toxic amounts.

### 3.2. Flour Characterization

#### 3.2.1. Water and Oil Absorption Index

The absorption properties of *S. madagascariensis* seed flour were observed through evaluating their WAI, OAI, solubility and swelling power. The WAI of flours has direct implications on its use in food products; typically, flours with high WAI are sought after for food products requiring high viscosities such as baked products, soups and gravies [33], while the OAI is important in facilitating the flavour enhancement of food products, as it promotes flavour retention and the increased palatability and mouth feel of food products. Additionally, it gives an indication of better reconstitution ability and good dough properties during the preparation of fat-related products, such as baked products [17]. The WAI and OAI of *S. madagascariensis* seed flour and the reference flours are indicated in Figure 1. The WAI of *S. madagascariensis* seed flour was found to be significantly different (*p* < 0.001) to the reference flours used. However, it was observed to be substantially high (compared to wheat, chickpea and tapioca flour), showing the second highest WAI just below that of coconut flour, thus indicating the seed flours potential as a thickening agent. The WAI is of particular importance during the processing of the flour, as it has direct effects on its pasting properties and represents the ability of a flour to associate with ‘under-water’ conditions (i.e., high temperatures) where such interactions are limited. The OAI of the seed flour showed similarities to brown rice flour, chickpea flour and tapioca flour. However, the OAI was substantially lower than that of wheat and coconut flour. Owing to the lack of starch in the seed flour, it is assumed that the WAI and OAI are attributed to the high fibre and sugar content of the flour, thus indicating the presence of cellulose and sucrose as these components are known for their water-holding capacities [34].

#### 3.2.2. Solubility and Swelling Power

Figure 2A,B, represents the S and SP patterns, respectively, of *S. madagascariensis* seed flour compared to those of reference flours. Water solubility is an indication of the soluble solid content of the flour, thus indicating the degree of starch degradation in relation to the level of maturity of the produce [35]. The seed flour showed a significantly (*p* < 0.001) high percentage of solubility across all the temperatures in comparison to the reference flours. Additionally, the seed flour showed a gradual increase in solubility with an increase in temperature, with a peak solubility at 100 °C. The SP of the flours is an indication of its water absorption properties [17]. The seed flour showed a maximum swelling power at 50 °C, which then gradually decreased with an increase in temperature. The swelling power of *S. madagascariensis* showed significant difference (*p* < 0.05) within this temperature range (60–100 °C). This is possibly due to the lack of starch granules within the seed flour, which hinder the increase in swelling. A similar pattern was observed for coconut flour, where no substantial increase in swelling occurred. The remaining reference flours (tapioca, wheat, chickpea and brown rice) all showed a gradual increase in SP with an increase in temperature. However, the overall swelling capacity of *S. madagascariensis* seed flour is similar to the maximum SP of the reference samples; suggesting that the seed flour possesses similar water absorption characteristics.

#### 3.2.3. FTIR

Flour properties of *S. madagascariensis* seed were analysed using FTIR and were compared to those of the reference flours, as depicted in the FTIR spectra below (Figure 3A,B). The seed flour of *S. madagascariensis* shows a similar FTIR spectral pattern to that of the reference flours, as it produces major peaks (although not very intense) within the same regions as the reference flours. These regions include N–H (3350–3310 cm^−1^), O–H (3000 cm^−1^), C = C (1650–1600 cm^−1^), N–O (1550–1500 cm^−1^), C–O (1150 cm^−1^) stretching and O–H (1440–1395 cm^−1^) and C = C (995–985 cm^−1^) bending groups (representing the following compound classes: secondary amine, carboxylic acid, conjugated alkene nitro compound, aliphatic ether, carboxylic acid and alkenes, respectively). The major peaks were identified through the use of IR spectroscopy correlation tables [36]. Regions between 1640 cm^−1^ and 3300 cm^−1^ are indicative of the moisture content as the water molecules possess a strong absorption within this range, due to the O–H stretching and H bending vibrations [37]. *S. madagascariensis* flour showed differences to most of the reference flours (excluding coconut flour) within the C–O stretching regions, of 1085–1050 cm^−1^ (arrow 1) and 1150–1085 cm^−1^ (arrow 2). These regions represent the primary alcohol group and the aliphatic ether group [36]. The FTIR spectrum shows that the seed flour of *S. madagascariensis* showed similarities to those of the commercial flours, suggesting that it may be used in similar food products.

#### 3.2.4. Pasting and Textural Properties

Table 3 compares the pasting properties and textural properties (penetration force) of *S. madagascariensis* seed flour to those of the reference flours used. The dashes in the gel texture column indicate data that could not be used, due to instrumental errors that occurred during the analysis. Flour pasting is defined as the process post-gelatinization, which includes the swelling and disruption of starch grains, and the release of molecular components. The presence of large amounts of fibrous material may hinder swelling and affect the viscosity of the sample during heating [38]. All the flours tested showed an increase in viscosity with increases in temperature. *S. madagascariensis* seed flour showed the second highest peak viscosity (3149 RVU) and the highest through the final and setback viscosities, as well as a relatively low breakdown viscosity. This was surprising as the flour had 0% starch (Table 1), implying there were no granules to facilitate swelling and gelatinization. Typically, the viscosities of the flour solutions were attributed to their starch content, as starch granule swelling and amylose leakage facilitate paste formation [39]. However, from the nutritional evaluation of the *S. madagascariensis* seed, the flour was found to contain no starch (Table 1), though with a high carbohydrate content (primarily composed of fibre and sugar). Thus, the high peak and final viscosities may be caused by the high sugar content of the flour, implying good potential use as a viscosity builder or stabilizer in food products similar to guar gum [40]. According to Pongsawatmanit [41], increases in sugar concentration (particularly sucrose) may increase both the peak and final viscosity of flour solutions, thus explaining the high viscosities previously mentioned. The pasting temperature of flour solutions is an indication of the point at which gelatinization is initiated, as this is the point at which sudden rises in viscosity occur. Thus, the pasting temperature provides a guideline of the minimum heat required to cook the flour to obtain the required viscosity [41]. *S. madagascariensis* seed flour showed a significantly low pasting temperature, indicating that less energy is required to gelatinize the flour.

Gel texture is a significant factor in food product development, as it indicates the ability of the gel to retain its structure and moisture content post processing [20]. The gel texture properties of *S. madagascariensis* seed flour and the reference flours are indicated in Table 3. The gel texture (hardness = firmness and penetration force) was significantly lower (*p* < 0.05) than that of the reference samples, contradicting expectations as it possesses a high peak and final viscosities. This indicates that the seed flour gel regressed much more easily than that of the reference flours. This could be due to the lack of starch, which is essential in maintaining the gels’ shape.

#### 3.2.5. Colour Properties

Colour profiling of *S. madagascariensis* was done according to the CIELAB scale and compared to that of the reference samples (Table 4). Additional colour evaluations were done, such as the chroma and FCI. Figure 4 below gives a visual depiction of the colour variation between the samples after being exposed to moisture and high temperatures. The figure shows pastes obtained from RVA in plastic sample holders. The degree of lightness (L *, redness (a *), yellowness (b *), chroma and FCI was significantly different (*p* < 0.05) to the reference flours used. The seed flour showed a higher degree of redness (a * = +0.9) and yellowness (b * = +12.7) compared to the reference samples, except for chickpea flour which had a yellowness value of +19.5, which may be due to the presence of the fruit pulp that was not completely removed from all of the seeds. Moreover, when comparing the colour of the pastes obtained from RVA (Figure 4), the seed flour paste was substantially darker than the reference flour pastes. This may be due to the non-enzymatic browning of the sample, facilitated by the exposure to high temperatures and moisture. The various reactions that may have led to the browning of the sample include the Maillard reaction, caramelization, chemical oxidation of phenols and maderisation [42]. However, further research is required to determine the particular reaction or reactions responsible for the increase in darkness of the flour after processing, as well as to determine the effects of these reactions on the antioxidant properties of the seed flour.

## 4. Conclusions

The seeds of *S. madagascariensis* have the potential to be a valuable food source due to the high amounts of fibre, sugar and certain minerals (especially K and Fe), which exceeds those of some commonly consumed products. The fibre content is of importance as this fraction may contribute greatly to the functioning of the gastrointestinal tract, while the high iron content may prove to be beneficial in combating iron deficiencies in South Africa. Although the seeds of *S. madagascariensis* hold valuable nutritional potential, it is crucial to develop food processing methods to reduce the strychnine content prior to their use in the food industry. Further research should be directed towards quantifying other phytochemicals, which might present anti-nutritional properties. The functional properties of *S. madagascariensis* seed flour may be beneficial in food product development as it possesses a high water absorption capacity, as indicated by its water absorption index and swelling power. Moreover, its high peak and final viscosities suggest that it may be used as a thickening agent in sauces or soups. The colour properties may be a hindrance, as the flour and the paste are substantially darker than the popular commercial flours. However, the consumer acceptability of varying flour colours is currently increasing. The commercialization of this underutilized fruit may also benefit the surrounding rural communities, as it provides entrepreneurial opportunities for the development of new food sources and the cultivation of this species. However, it is essential to rule out any toxicity hazard of the seeds’ prior use in the food industry.

## Figures and Tables

**Figure 1 foods-09-01060-f001:**
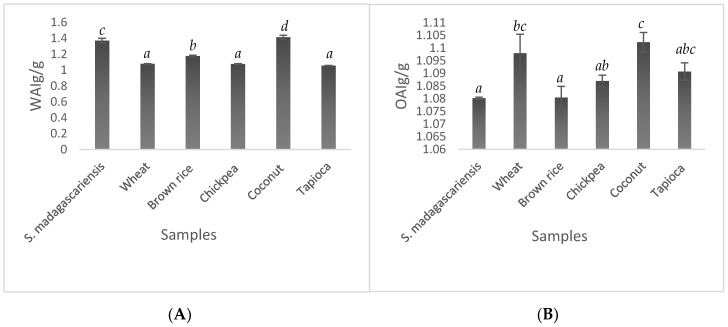
Water (**A**) and oil (**B**) absorption index of *Strychnos madagascariensis* seed flour compared to five commercial flours. The error bars indicate the standard deviations of each of the samples, with Least Significant Difference values of 0.0248 and 0.00737 for water absorption index (WAI) and oil absorption index (OAI), respectively. Bars with the same letters above them show no significant differences (*p* < 0.001).

**Figure 2 foods-09-01060-f002:**
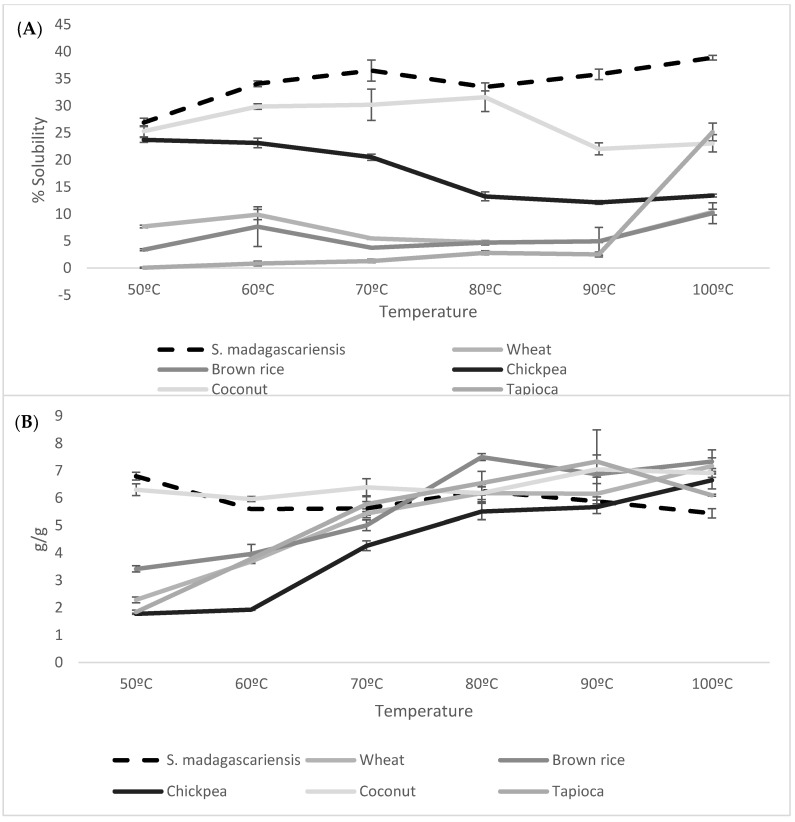
Solubility (**A**) and swelling power (**B**) of *Strychnos madagascariensis* seed compared to commercial flours, at varying temperatures. The error bars indicate the LSD values for each of the samples.

**Figure 3 foods-09-01060-f003:**
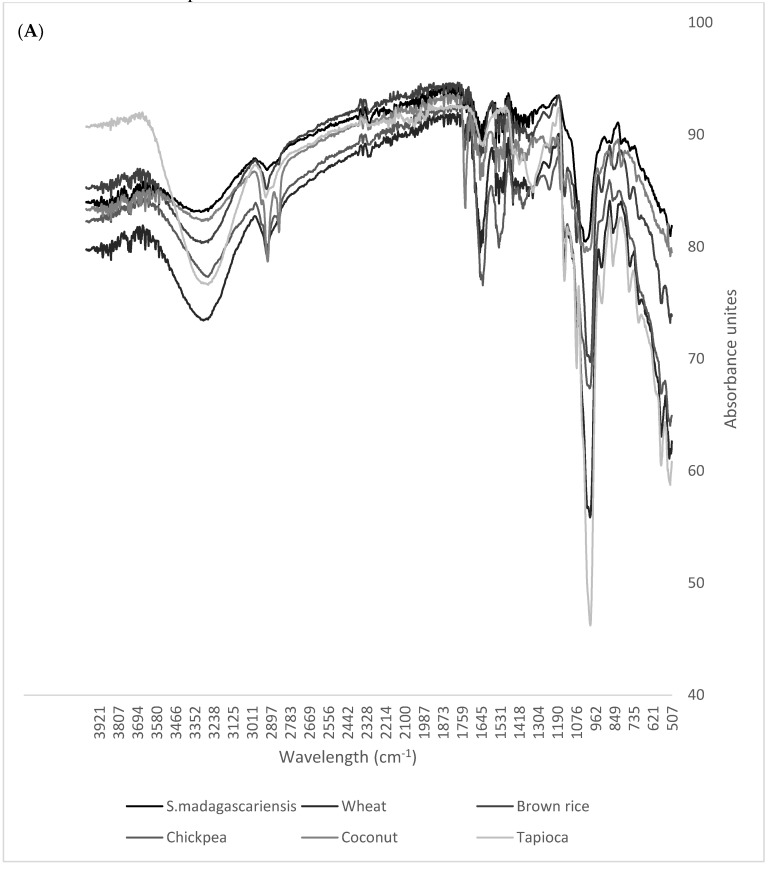
(**A**) Superimposed FTIR spectra of the *S. madagascariensis* seed flour and the reference flours (wheat, brown rice, chickpea, coconut and tapioca), and (**B**) the region where the differences are observed in the *S. madagascariensis* seed flour.

**Figure 4 foods-09-01060-f004:**
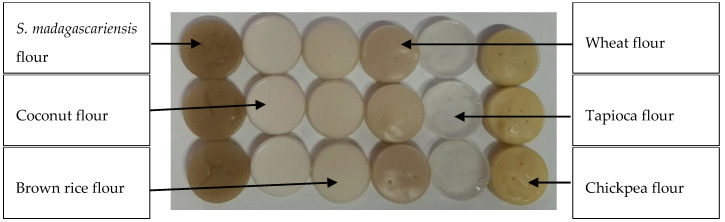
Sample pastes obtained from the rapid visco analyser (RVA) indicating the colour variations between the *S. madagascariensis* seed flour and the reference flours, for which the pastes were obtained by creating a suspension of flour and water (14% *w*/*w*), heating to a maximum temperature of 91 °C and allowing to cool for 2 h at 25 °C.

**Table 1 foods-09-01060-t001:** Nutritional composition and strychnine content of *S. madagascariensis* seed flour (per 100 g) and the flours’ percentage contribution to NRVs.

Nutritional Category	Composition	NRV	Percentage Contribution to NRVs
**Calories**	189.03 kcal ± 0.00	2000 kcal	9.45%
**Total Fat**	0.95 g ± 0.00	44.44–77.78 g	2.14%
**Total Carbohydrates** **Total fibre** **Sugar** **Starch**	89.85 g 53 g ± 3.65 41 g ± 4.69 0 g ± 0.00	225–300 g 25 g <50 g 150–225 g	34.94% 200% 82% 0%
**Protein**	8.27 g ± 0.13	49.8 g	16.61%
**Strychnine**	0.08% ± 0.04	N/A	N/A
**Ca**	148.03 mg ± 6.00	700 mg/day	21.15%
**Mg**	79.25 mg ± 1.00	285 mg/day	27.80%
**K**	594.36 mg ± 4.00	3500 mg/day	16.98%
**P**	94.20 mg ± 2.00	2400 mg/day	3.93%
**Na**	16.45 mg ± 1.00	1600 mg/day	1.03%
**Zn**	1.36 mg ± 0.02	42 mg/day	3.24%
**Mn**	9.86 mg ± 0.1	12.2 mg/day	80.82%
**Fe**	15.78 mg ± 0.4	6.7 mg/day (males) 11.4 mg/day (females)	235.52% 138.42%
**Cu**	0.72 mg ± 0.01	10 mg/day	7.2%

Nutrient reference values (NRVs) are a set of values that are used as a reference point for the daily intake of micro and macronutrients, which are required to maintain a healthy and balanced diet [14]. Table 1 above indicates the NRVs for both micro and macronutrients as well as the percentage contributions of *S. madagascariensis* seeds to these values. From this table, it can be seen that the total fibre, sugar, Mn and Fe contribute greatly towards these daily recommended values (200, 82, 81 and 236/138%, respectively). This further highlights the potential of *S. madagascariensis* seeds as a food alternative or supplement, aiming to alleviate food insecurity and mineral deficiencies.

**Table 2 foods-09-01060-t002:** Volatile metabolic compounds identified in *S. madagascariensis* seed flour in reference to the other flours.

Compound	Retention Time (s)	Mass (BPI)	*S. madagascariensis*	Wheat	Brown rice	Chickpea	Coconut	Tapioca
Pyrazole	179.858	43.0179	X	X		X		
Vinyl acrylate	204.113	55.018	X				X	
Ethylene acetal	189.010	73.0284	X				X	
Furfuryl alcohol	214.603	98.0364	X					
dl-Alanine ethyl ester	222.354	44.0258	X					
Methoxy-phenyl-oxime	229.186	133.0137	X	X	X			X
Cyclotene	309.017	112.0519	X					
Furaneol	342.05	57.0336	X					
Butanoic acid	240.122	112.0519	X					
Methylcyclopentenolone	309.017	112.0519	X					
Hydratropaldehyde	376.572	105.0699	X					
Silicon tetrafluoride	456.775	85.0284	X					
2-Acetoxy-5-hydroxyacetophenone	488.172	137.0234	X					
Styrene glycol	491.52	107.0493	X					
Trifluoromethylcinnamic acid	726.178	199.0601	X					
a-Dnp-L-arginine	913.886	69.0955	X					
Benzenepropanoic acid	496.505	107.0604	X					
Cyclo(leucyloprolyl)	1043.49	70.0653	X					
Quinoline	502.308	138.0676	X					

BPI = base peak intensity. X indicates the presence of the compound.

**Table 3 foods-09-01060-t003:** Pasting properties of the *Strychnos madagascariensis* seed flour compared to the five reference flours.

Flour Type	Peak Viscosity (RVU)	Trough Viscosity (RVU)	Breakdown Viscosity (RVU)	Final Viscosity (RVU)	Setback Viscosity (RVU)	Peak Time (min)	Pasting Temp (°C)	Gel Texture (N)
*S. madagascariensis*	37788 ± 2016 ^b^	20580 ± 96 ^c^	17208 ± 1944 ^b^	62928 ± 552 ^e^	42348 ± 492 ^e^	2 ± 0.1 ^a^	50 ± 0.1 ^a^	3 ± 0.4 ^a^
Coconut	7548 ± 1008 ^d^	3552 ± 864 ^a^	3984 ± 600 ^a^	12804 ± 3240 ^a^	9240 ± 2508 ^b^	5 ± 0.1 ^c^	93 ± 0.4 ^f^	-
Brown rice	18924 ± 888 ^c^	15000 ± 432 ^b^	3924 ± 1116 ^a^	34416 ± 408 ^c^	19416 ± 624 ^c^	6 ± 0.3 ^cd^	86 ± 0.5 ^e^	14 ± 0.7 ^d^
Wheat	33540 ± 744 ^b^	13980 ± 252 ^b^	19560 ± 492 ^b^	31848 ± 1032 ^c^	17868 ± 828 ^c^	5 ± 0 ^cd^	67 ± 0.5 ^b^	11 ± 1.6 ^c^
Tapioca	78228 ± 5796 ^a^	14124 ± 2016 ^b^	64092 ± 4944 ^c^	40872 ± 3804 ^d^	26736 ± 2496 ^d^	3 ± 0.1 ^b^	70 ± 0.5 ^c^	-
Chickpea	18132 ± 516 ^c^	15744 ± 528 ^b^	2388 ± 84 ^a^	20184 ± 600 ^b^	4440 ± 72 ^a^	6 ± 0.2 ^d^	78 ± 0.4 ^d^	5 ± 0.7 ^b^

Means accompanied with the same letter in a given column show no significant differences (*p* < 0.05).

**Table 4 foods-09-01060-t004:** Colour properties of the *Strychnos madagascariensis* seed flour compared to the five reference flours.

Sample	L*	a*	b*	Chroma	FCI
*S. madagascariensis*	81 ± 0.3 ^a^	0.9 ± 0.1 ^e^	12.7 ± 0.6 ^c^	12.7 ± 0.6 ^c^	68.5 ± 0.6 ^b^
Coconut	86 ± 0.2 ^c^	0.2 ± 0.1 ^b^	11.3 ± 0.4 ^b^	11.3 ± 0.4 ^b^	74.9 ± 0.5 ^c^
Brown rice	84 ± 0.3 ^b^	0.5 ± 0.1 ^c^	10.6 ± 0.3 ^b^	10.6 ± 0.3 ^b^	74.3 ± 0.1 ^c^
Wheat	87 ± 0.3 ^d^	0.6 ± 0.1 ^d^	10.7 ± 0.2 ^b^	10.7 ± 0.2 ^b^	76.8 ± 0.5 ^d^
Tapioca	93 ± 0.2 ^e^	-0.2 ± 0.1 ^a^	3.8 ± 0.2 ^a^	3.8 ± 0.2 ^a^	89.3 ± 0.3 ^e^
Chickpea	86 ± 0.3 ^c^	1.1 ± 0.1 ^f^	19.5 ± 0.2 ^d^	19.5 ± 0.2 ^d^	66.6 ± 0.1 ^a^

Means accompanied by the same letter in a given column are not significantly different (*p* < 0.05). L* (white = 100 and black = 0), a* (positive = red and negative = green), b* (positive = yellow and negative = blue). FCI – Flour Colour Intensity.

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
