# Peer review of "Nutritional and Physicochemical Characterization of Strychnos madagascariensis Poir (Black Monkey Orange) Seeds as a Potential Food Source"

_foods, 2020, doi:10.3390/foods9081060_

Round 1
Reviewer 1 Report
This manuscript has the some strengths and weaknesses, that I explain below divided in 4 points:
1.- It reflects a very complete analysis of the seeds of Strychnos madagascariensis, including chemical composition (using sound and appropriate methodologies) and also physical behaviour regarding texture analysis.
2.- From the point of view of the experimental work, as understood from the manuscript, only one sample has been analysed. Natural variability is known to be wide, so to provide a more real information about this product, a wider sampling should have been considered, including seeds from different trees, growing in different places, different soils, climates, seasons...Otherwise, the data are not representative enough.
3.- Minor points regarding terminology are:
- The inclusion of the author of the species in the title, abstract, and in the text first time is written.
- Do not use a trademark for samples, as it is made for "Snowflake cake wheat flour", but a descriptive name of what it is (gluten enriched wheat flour?).
- Available carbohydrates are referred as sugars throughout the text, while they comprise starch + sugars. Use a more precise term to refer to them. Line 365-366: how do authors know that there is a lack of starch in these seeds?
- Lines 201: Not all the fibre components are prebiotics, just some of them. Do not apply this term to the whole amount of fibre found.
- Line 203: Some fibre components promote minerals absorption; others impair them. Do not refer only to the positive effect if a detailed analysis of fibre fraction has not been done.
4.- The most important point in this manuscript is safety of the analysed seeds:
Food security cannot be improved compromising food safety, and in this case, information provided by literature, as well as that provided in the present study are enough to avoid any mention about food uses of these seeds.
The reasons for that are:
- In lines 56-57: "However, there has been contradictory claims surrounding the presence of this alkaloid as Govender [5] and Motlhanka et al. [7] reported that the seeds are edible and do not contain strychnine".
Reference 5, subtly, but not clearly, suggest some use of the seeds; and no mention to its content of strychnine is made. Ref. 7 only mentions that "apparently" is has no alkaloids, with no references nor analytical data. The statements in these references are not supporting this sentence so far.
- On the contrary, results of this manuscript, demonstrate the presence of strychnine in seeds of S. madascariensis. Lines 235-237: "The strychnine content of S. madagascariensis seeds (0.08%) was found to be substantially lower than that of S. nux-vomica (~ 1.1%) [30], suggesting that the toxic effects might not necessarily be of concern, as the alkaloid is present in low quantities". This is not supported by the results: we only know the levels, any fact make us conclude that there are no concern. Just being lower than in S. nux-vomica, does not mean it is low, and neither is a guarantee of safety.
Furthemore, just one sample has been analysed. Considering natural variability, how do authors know that there are not higher amounts in other samples of the same species?
- Previous reports in different sources, from Administrations or scientific publications (including references 30-33 of this manuscript), reports variable lethal oral doses of strychnine, ranging from 30 to 120 mg for adults. Also, in Philippe et al. (2004) very serious adverse effects are reported after 5-30 mg ingestion (Philippe, G.; Angenot, L; Tits, M.; Frédérich, M. 2004. About the toxicity of some Strychnos species and their alkaloids. Toxicon, 44(4): 405-416).
Given the amount found in this manuscript (0.08 % = 80 mg/100 g), This mean that just 40 g of seed flour could be lethal for an adult (less for children, given the lower body weight). This amount are easily edible.
- Lines 237-238: " the processing of these seeds may further reduce or eliminate the strychnine content", and lines "241-244: "Although the seeds are the main target, the leaves, bark and fruit are utilized for herbal remedies as well after they have been processed (boiled or cooked) in order to reduce the quantity of strychnine [31,32]".
Ref. 31, apply sand heating to seeds of S. nux-vomica, founding heat stability of alkaloids strychnine and brucin; then several extractions using different solvents could reduce strychnine amount. This procedure was devoted to develop drugs enriched in brucin. However, whether it is applicable to food uses is questionable. Anyway, if authors consider that this would be an alternative to obtain detoxified seeds, this would be a justification for the paper, but always giving references to the efficiency of extraction, and the characteristics of the obtained product, not just as a hypothesis, but as a real possibility.
On the other hand, Ref. 32 present some warnings about poisoning with herbal drugs made of S. nux-vomica, even if they were processed for detoxification, so this process must not only suggest, but guarantee the safety of the products.
- The manuscript talks about a traditional use, but no reference to ethnobotanic data are given about a real and safe traditional long-time consumption of these seeds. Given the seriousness of strychnine poisoning, and the absence of a antidote, one think is that somebody, on his own responsibility, eat some plant product that may, or may not, cause adverse effects, and other is that researchers encourage it. Scientists must be especially prudent in our recommendations, which may be quickly put out of context, and eventually reach the edge of toxicity.
For all that, I consider that if accepted, this manuscript has to be rewritten with the focus on two possibilities:
- Present the work as just a chemical analysis of S. madagascariensis seeds, to improve scientific knowledge about their composition (see the revision of Philippe et al., 2004: no data about this species is recorded, so this a new information), but never recommending their use as food. With this focus, it would be suitable for other Journal (not Foods).
- If authors want to keep the focus in foods, the work should be presented as a food safety warning for the potential use of these seeds in human nutrition, not to encourage it. In this way, and with the referred improvements, or others suggested by other reviewers, it could be suitable for this Journal, but changing title, aim, conclusions...
Author Response
Please use the Word file attached

Reviewer 2 Report
The manuscript “Nutritional and physicochemical characterization of Strychnos madagascariensis (black monkey orange) seeds for food security” by K. K. van Rayne, O. A. Adebo, O. Wokadala, and N. Z. Ngobese deals with the very important and current problem of the food scarcity and the food low nutritional value. The Authors investigated new plant sources which might be of interest to food production. Black monkey orange, which seeds have been the object of their study, is the plant not well known outside the region of its occurrence, therefore I think that the manuscript is a very valuable contribution to the knowledge of plant natural food products. However, in my opinion the paper is lacking a very important issue: seeds of this plant contain strychnine, the poison. The Authors compare the content of this substance in seeds of black monkey orange with that found in S. nux-vomica, but the latter is used mainly as medicine, so the declaration, that in S. madagascariensis this content is lower and it MIGHT be reduced by processing is not satisfactory. The food safety is extremely important and first of all the strychnine content in the seeds should be positively and precisely examined (the data given in Table 1 seems to be rather approximate) and if it allow to use the seed flour for food production. I agree that the nutritional value of S. madagascariensis seeds is very high, but if the strychnine content turns out too high, it cancels the possibility of its utilization.
From practical point of view, it would be also interesting to know, what part of the fruit are seeds (mass percentage), meaning whether obtaining seed flour is economical. Besides, if its taste, probably bitter, would be acceptable to consumers?
My other remarks are, as follows:
- line 12 - “South Africa” should be added to address
- word repetition in lines 17 and 18 (“characterizing/characterized”), the same in line 322 (“flour”), and in lines 256 and 258 (“properties”)
- line 20 - should be “contained”
- line 57 – wrong reference number (Motlhanka et al. is not [7])
- Table 1 should not be divided into two pages
- Table 1: why only strychnine content is given in %?
- line 247 - either “the’ or “its” should be deleted
- Section 3.1.2 GC-MS analysis
This part of the manuscript should be read carefully and corrected. It is written rather poorly. There are some grammar mistakes (for instance in lines 253-254 should be “the compounds (…) include”). There are also incorrect statements (pyrazole (l. 254) is rather a compound, not a group) or obvious ones: vinyl acrylate (l. 258) falls under (meaning: belongs to?) acrylates. Besides, some information given on compounds detected in seed flour (i.e. their application to photopolymerization or to nylon production) have nothing to do with food or nutrition problems, the main topics of the manuscript and should be deleted. Instead, it would be worth mentioning, if these compounds are not potentially harmful or toxic to humans, as it was said of some of them.
- Table 2 – the title of the table: the word “signature” (line 268) is incorrect; in line 269 should be “in reference to…”
The compounds names and the column headings should not be divided automatically by computer program
- line 272 - “the interaction of S. madagascariensis seed flour” - interaction with what?
- line 285 – what do you mean by saying that WAI represents the ability (…) “under conditions where such is limited” ?
- Section 3.2.3 FTIR
Also this section should be corrected. The wavenumbers of particular bands characteristic for mentioned chemical groups (line 326) should be given. The IR region (1640 cm-1 – 3300 cm-1) characterized in lines 329-331 is not shown in the spectrum. The region 1085-1050 cm-1 is included in the region 1085-1150 cm-1 (lines 333), so it is difficult to see what bands are of interest. The arrow B points to the region, where the differences between spectra are negligible. And finally, there is no comment on the FTIR results.
- Table 3 - I would suggest presenting data as a figure (bars or plots)
- line 375 and Figure 4 – the particular conditions (moisture and high(?) temperature) of pastes preparation should be given.
Author Response
Please use the Word file attached

Reviewer 3 Report
Manuscript: Nutritional and physicochemical….black monkey orange seeds for food industry’ reports on the nutritional composition and characterization of functional properties of Strychnos madagascariensis seeds. This in my opinion, is a useful study focusing on the potential for use of black monkey orange seeds by the food industry in Africa. What worries me is (i) the deleterious potential of strychnine in the seeds and (ii) the toxicity of metabolites listed in Table 2 i.e. methylcyclopentenolone, silicon tetrafluoride, etc ! Before reaching the conclusion that seeds are edible, the authors should make sure that (i) the product is absolutely safe for use and (ii) it does not introduce any undesirable sensory attributes to foods prepared using it. Is the strychnine content determined (0.08 %) negligible in terms of deleterious effect ? Would it have been a good idea to test its functional properties attempting to prepare i.e. bread, in which case the authors would have seen the effect of thermal treatment on the strychnine content of the end product ?
My detailed comments follow the text sequence
l.153: change ‘less’ to ‘non’
l.130-176: all these properties determined, would have been tested in an application i.e. bread preparation which would be included in the study (given, of course the addressing of the safety problem first !)
l.205: the seeds are high in energy; however fiber (>50 % w/w) cannot be used as a source of energy in the human body, thus the statement in incorrect.
l.238-239: the authors, correctly state that ‘further investigation is required to evaluate the strychnine content after processing’. This issue along with the more general toxicity issue of strychnine should be addressed before any recommendations can be made to the food industry on the use of black monkey orange seeds.
Table 2: some of the metabolites identified are toxic (see my general comment). If there is no quantification of such compounds, how can one recommend the seeds’ use by the food industry ?
l.401-402: this is simply a hypothesis that should be proven !
Based on the above, the least the authors can do is to address all above issues in their revised text.
Author Response
Please use the Word file attached

Round 2
Reviewer 1 Report
The manuscript has been improved in agreement with the comments previously made.
However, there are some points that still need improving:
- As previously explained authors should be even more prudent in recommending the food consumption of a product containing such a harmful compound. They have decided to maintain the focus on the study on the use this fruit as a food, but in this case, any allusion to "nutritional" should be changed to "nutritional potential" . It is necessary to transmit that that the fruit is evaluated because it may have interesting nutrients but it cannot be recommended as a food until its safety is clearly and scientifically demonstrated. This is well explained some places of the manuscript, but still some allusions giving the idea that it can already be recommended as food, are present in some places. The most important ones are in the title and aims; please include the word "potential" in these and other places of the text.
- Lines 248-252: "The strychnine content of S. madagascariensis seeds (0.08%) was found to be substantially lower than that of S. nux-vomica (~ 1.1%) [30], suggesting that the toxic effects might not necessarily significant". AS indicated in the previous revision, these values do not suggest this. The only thing that demonstrates is that S. madascariensis has lower strychnine content than S nux-vomica. To demonstrate that there is no risk, toxicological studies using S. madascriensis should be done; or the values should be compared with the toxic dose of strychnine for humans. This comparison does not suggest absence of toxicity, so please delete the last part of the sentence, and just leave the fact that values are lower.
- Line 59-61:"Govender [5] and Motlhanka et al. [76] reported that the seeds are consumed. This has not been proven and is not supported by any analytical data". To demonstrate that seeds are consumed, no analytical (laboratory) data are needed, unless authors mean statistical analysis of ethnobotanical data. I suggest to change "analytical" by "ethnobotanical"
- Lines 253-270: please review this part regarding the following. Authors explain the possibility of removing strychnine by processing citing studies made for the development of herbal drugs or other similar products. Processing for the obtention of extracts to be used in drugs are quite different than those suitable for the production of foods, so for this purpose the studies should be based on those already done, but just assay the efficiency in removing strychnine by those process suitable for food production. Please be sure that this point is clear in this paragraph.
Author Response
Please see the Word document attached.

Reviewer 3 Report
l.368: change 'shown similarities to that' to 'showed similarities to those'. There may be other grammatical errors in the revised text. See that the text is proof read by a colleague proficient in English.
Author Response
Please see the Word document attached.
